# Nordic Walking and Free Walking Improve the Quality of Life, Cognitive Function, and Depressive Symptoms in Individuals with Parkinson’s Disease: A Randomized Clinical Trial

**DOI:** 10.3390/jfmk5040082

**Published:** 2020-11-10

**Authors:** Elren Passos-Monteiro, Felipe B. Schuch, Leandro T. Franzoni, Alberito R. Carvalho, Natalia A. Gomeñuka, Marindia Becker, Carlos R. M. Rieder, Alexandro Andrade, Flávia G. Martinez, Aline S. Pagnussat, Leonardo A. Peyré-Tartaruga

**Affiliations:** 1Exercise Research Laboratory, Universidade Federal do Rio Grande do Sul (UFRGS), Porto Alegre 90040-060, RS, Brazil; elrenpm@ufcspa.edu.br (E.P.-M.); alberitorodrigo@gmail.com (A.R.C.); natalia.gomenuka@gmail.com (N.A.G.); marindia.becker@gmail.com (M.B.); fla.gmartinez@gmail.com (F.G.M.); leonardo.tartaruga@ufrgs.br (L.A.P.-T.); 2Health Sciences Graduate Program, Universidade Federal de Ciências da Saúde de Porto Alegre (UFCSPA), Porto Alegre 90040-060, RS, Brazil; alinespagnussat@gmail.com; 3Human Movement Sciences Graduate Program, School of Physical Education, Universidade Federal do Pará (UFPA), Castanhal 66075-110, PA, Brazil; 4Department of Sports Methods and Techniques, Universidade Federal de Santa Maria (UFSM), Santa Maria 97105-900, RN, Brazil; 5Postgraduate Program in Medical Science, Division of Cardiology, Hospital de Clínicas de Porto Alegre, Universidade Federal do Rio Grande do Sul (UFRGS), Porto Alegre 90040-060, RS, Brazil; franzoni.esef@gmail.com; 6Physical Therapy College, Universidade Estadual do Oeste do Paraná (UNIOESTE), Cascavel 85819-170, PA, Brazil; 7Departamento de Investigación de la Facultad de Ciencias de la Salud, Universidad Católica de las Misiones, Posadas (UCAMI), Misiones N3300, Argentina; 8Movement Disorders Clinics, Division of Neurology, Universidade Federal de Ciências da Saúde de Porto Alegre (UFCSPA), Porto Alegre 90050-170, RS, Brazil; carlos.rieder@gmail.com; 9Laboratory of Psychology of Sport and Exercise, Department of Physical Education, Center of Health Sciences and Sports, Santa Catarina State University (UDESC), Florianópolis 89223-100, SC, Brazil; alexandro.andrade@udesc.br; 10Rehabilitation Sciences Graduate Program, Movement Analysis and Neurological Rehabilitation Laboratory, Universidade Federal de Ciências da Saúde de Porto Alegre (UFSCPA), Porto Alegre 90050-170, RS, Brazil; 11Human Movement Sciences Graduate Program, Universidade Federal do Rio Grande do Sul, Porto Alegre 90040-060, RS, Brazil

**Keywords:** quality of life, cognitive, depression, neurodegenerative diseases, Parkinsonian disorders, exercise therapy

## Abstract

Nordic walking’s (NW) degree of effectiveness regarding health-related parameters in people with Parkinson’s Disease (PD) is a subject of debate. While NW seems to improve functionality, a clear non-motor benefit has not been demonstrated. The aim of this randomized controlled trial was to compare the effects of 9-week NW and free walking (FW) training programs on quality of life, cognitive function, and depressive symptoms in individuals with PD. Thirty-three people with PD, (Hoehn and Yahr 1–4) were randomized into two groups: NW (*n* = 16) and FW (*n* = 17). We analyzed quality of life, cognitive function, depressive symptoms, and motor symptoms. Significant improvements were found in the overall, physical, psychological, social participation, and intimacy domains of quality of life, as well as in cognitive function and depressive symptoms for both groups. Only the NW group showed improvement in the autonomy domain. Individuals with PD had a similar enhancement of non-motor symptoms after walking training, with or without poles. However, the NW group showed a more significant improvement in the autonomy domain, strengthening the applied and clinical potential of NW in people with PD. Future studies are needed to determine the efficacy of walking training without poles in subjects with PD.

## 1. Introduction

Changes in the central nervous system resulting from Parkinson’s disease (PD) lead to the manifestation of non-motor symptoms, such as cognitive deficits and increased depressive symptoms [1,2]. The severity of non-motor and motor symptoms, in turn, is causally related to deficits in quality of life (QoL) [3,4]. Furthermore, some studies show the influence of sedentary time on factors associated with quality of life in people with PD [5].

In addition to drug therapy, exercise has been considered as a therapeutic tool for treating depression and improving QoL in people with PD [5,6]. In PD, compelling evidence suggests that exercise can improve non-motor symptoms such as cognitive deficits, depression, and apathy [7,8,9]. However, it is not clear whether exercise can improve QoL.

Free walking is the standard physical activity recommended for this population. However, Nordic walking (walking using two sticks) has been indicated, mostly because of its greater motor complexity requiring the activation of other cortical regions such as the premotor cortex [10,11]. It is proposed that the higher complexity of neural circuitry involved in Nordic walking may contribute the promotion of an improvement in cognitive deficits [7,12,13].

From a mechanistic standpoint, enhanced cognitive function implies mitigating not just movement disorders but also non-motor symptoms including autonomic dysfunction, sleep disorders, and pain [4,5,14]. Although positive motor and clinical adaptation using Nordic walking in patients with PD have been shown [10,12,15,16], to the best of our knowledge no data have been reported about the effects of Nordic walking in comparison to a free walking training program on QoL and non-motor symptoms in individuals with PD.

These training programs might induce adaptations that might influence health-related QoL, because non-motor symptoms have, collectively, a greater impact on QoL than motor symptoms [17]. We hypothesized that Nordic walking can be more effective than free walking in improving QoL and non-motor symptoms in PD because of the stimulation of mechanoreceptors provoked by excitatory stimuli from using poles, leading to an increased improvement in these outcomes. Thus, this study aimed to examine the effects of nine-week Nordic walking and free walking training programs on QoL, non-motor symptoms (cognitive function, depressive symptoms), and motor symptoms in people with PD.

## 2. Materials and Methods

This study is a superiority randomized clinical trial in parallel, single-blinded (evaluator-blinded), conducted according to the Consolidated Standards of Reporting Trials (CONSORT) recommendations. No changes in methods after trial commencement were made. The study was approved by the ethics committee of Hospital de Clínicas de Porto Alegre (HCPA) (Clinics Hospital of Porto Alegre) (ref no. 24595713.4.0000.5327) and was registered at ClinicalTrials.gov (clinical trial identifier: NCT03355521). The study was performed in accordance with the 1964 Helsinki declaration and its later amendments. All participants read and signed an informed consent form before starting their participation in the study. All participants were recruited from the Neurology ambulatory of HCPA, in Porto Alegre, Rio Grande do Sul, Brazil. After the initial contact by telephone, patients were invited to participate in the study according to eligibility criteria. Assessments and training sessions were performed at the Escola de Educação Física, Fisioterapia e Dança (ESEFID) da Universidade Federal do Rio Grande do Sul (UFRGS) (School of Physical Education, Physiotherapy and Dance of the Federal University of Rio Grande do Sul, Figure 1).

### 2.1. Sample and Randomization

We aimed to perform a clinical trial testing the superiority of a relatively new endurance modality (Nordic walking) against a well-known endurance activity (free walking) that has already shown positive results in the literature. We chose to compare the superiority effects of two interventions, i.e., Nordic walking and free walking, for ethical reasons. Physical exercise is extremely important in maintaining quality of life and improving symptoms in people with PD. The sample size calculation showed that 30 patients would be sufficient (15 in each group) for a power of 90% at an α = 0.05 and a correlation coefficient of 0.9 for the variables, based on standard deviations and the differences between the means of a study with QoL outcomes [17]. Considering possible withdrawals, the sample size was increased to 34 patients (*n* = 34). Patients who agreed to participate in the protocol were assigned a number and randomly allocated into one of the two groups using an online method to randomize using software (www.randomization.com). Allocation was concealed using sequentially numbered and sealed opaque envelopes that were opened only when each participant was admitted to the study. An independent assessor controlled the implementation and the sequence of randomization, with allocation index of 1:1.

Eligibility criteria were: (1) having a clinical diagnosis of idiopathic PD, according to Parkinson’s Disease Society Brain Bank criteria, by a neurologist; (2) PD severity ranging from 1 to 4 on the Hoehn and Yahr Scale (H&Y); (3) currently using of pharmacological treatment for PD; (4) aged 50 years or more; (5) being sedentary or not engaging in any type of physical activity for at least six months [18]; (6) not having undergone recent surgery or Deep Brain Stimulation (DBS); (7) and not having any clinical condition that limited or contraindicated the practice of exercise. Participants were informed about the procedures and phases of the study and signed an informed consent form. Exclusion criteria were having undergone any surgical procedures in the previous six months; having some heart disease or uncontrolled blood pressure, having had a myocardial infarction in the previous 12 months, using a pacemaker; having had a stroke or having other neurological diseases such as dementia; or having acute pain or prostheses in the lower and upper limbs that would make walking impossible. The HCPA online medical record assessed all information.

Following baseline testing, participants were randomly allocated into two training groups: Nordic walking and free walking. Interventions were conducted for both groups with two sessions per week, 35 min per session, progressing to up to 60 min total in the last training cycle, totaling nine weeks, always during the ON period of the dopaminergic medication cycle. Interventions were supervised by physical education professionals and physiotherapists, who provided immediate physical assistance when necessary during the training session.

### 2.2. Outcomes

The primary outcome of the present study was QoL. QoL has been extensively utilized for purposes of assessing the level of satisfaction with activities and subjective feelings about life conditions and has demonstrated good levels of reproducibility and validity [19]. Secondary outcomes were cognitive function, depressive symptoms, motor symptoms, and severity of the disease. Outcome measurements were performed at two different times (Pre: initial evaluation before training and Post: evaluation after training). All assessments were conducted in the “ON” period of the medication cycle, between one and three hours after anti-Parkinsonian drug intake.

### 2.3. Assessments

Quality of life outcomes and non-motor parameters (cognitive function and depressive symptoms) were compared between Nordic walking and free walking, in terms of time (pre- and post-intervention), and group × time interaction. QoL was assessed with the following instruments: the World Health Organization Instrument for Quality of Life Assessment questionnaire (WHOQOL-BREF domains: physical, psychological, social relationships, environment, and general quality of life; and WHOQOL-OLD domains: sensory abilities, autonomy, ‘Past, Present and Future Activities’, social participation, death and dying, intimacy, and general quality of life) [19]. Non-motor symptoms (cognitive function and depressive symptoms) were assessed. The Montreal Cognitive Assessment (MoCA), is a brief screening tool for mild cognitive impairment. This evaluation assesses different cognitive domains and investigates the individual’s abilities in the following areas: attention and concentration, executive functions, memory, language, visuo-constructive skills, conceptualization, calculation, and orientation. The total score of the MoCA is 30 points. A score of 26, or more, is considered normal, and a score of less than 26 is considered as cognitive impairment. Depressive symptoms were measured by the Geriatric Depression Scale–15 items (GDS-15). The scale consists of 15 dichotomous questions in which participants are asked to answer yes or no about how they felt over the past week (for instance, “Does the patient feel that their life is empty?” “Does the patient feel that their situation is hopeless?”). Scores range from 0 to 15, with higher scores indicating more depressive symptoms.

Motor symptoms and severity of PD were evaluated using the Unified Parkinson’s disease rating Scale-III (UPDRS-III) and H&Y, respectively [4,20]. Anthropometric data such as body mass and height were collected in both tests for the characterization of the sample, using a scientific balance (FILIZOLA, São Paulo, Brazil). Two assessors conducted the mass and height assessments to avoid possible faults.

### 2.4. Exercise Training

Both training programs, i.e., Nordic walking and free walking, consisted of three parts: (1) joint mobilization + warm-up with a three-minute free walk at a self-selected walking speed (SSWS); (2) main part—specific training of the group; (3) cool-down. The Nordic walking group used poles recommended for the correct practice (Newfeel^®^–Decathlon, Nordic Walker, China), and learning of NW technique was based on the Four Support System of Nordic Walking: Brazil Locomotion Method (Figure 2).

The NW and FW training protocol progressed over time until reaching 60 min in the last training cycle. Every three sessions there was a 40-min regenerative session with walking at a SSWS, stretching with poles (for the NW group), and body relaxation techniques. The training protocol consisted of three parts: (1) joint mobilization and warm-up with a three-minute free walk at a SSWS (eight min); (2) main part–specific training (35 to 50 min): continuous or interval walking on different terrains (i.e., track, grass and sand), with two to five minutes of rest in predetermined times. The training protocol was standardized among all subjects. The difference between the workout sessions in this part of the session was the use or not of poles during walking; (3) cool-down with stretching, and core and breathing exercises (five minutes).

Training was individually prescribed according to the maximum distance performed in 6-min walking test for determining the volume of training. Heart rate was used for determining the intensity of training based on the mathematical model of Tanaka [18]. We used a heart rate monitor, FT4 model (Polar Electro Oy, Kempele, Finland) attached to the chest at the level of the xiphoid process to control the progression of intensity. Training cycles ranged from 60–80% of heart rate reserve. Additionally, we used the Borg ratio of perceived exertion (Borg RPE) for training intensity control; intensity ranged between 13–17 and matched the intensity of the heart rate. The total time of the sessions was determined every three sessions in accordance with the training cycles. The periodization method is given in detail in Monteiro et al. [12].

To facilitate the learning process of the Nordic walking movement technique, we used the rhythmic verbal command, “1,2,3,4”. Patients were instructed to count at this rate and to imagine the time by step frequency. To be fair, it was proposed to the participants in the free walking group that they could experience NW after the nine-week free walking intervention. Two videos showing training examples are given in Appendix A.

### 2.5. Statistical Analysis

Data are presented as mean and standard deviations or 95% confidence interval (CI 95%) for continuous measures. Baseline characteristics were compared by using one-way analysis of variance (ANOVA). Intent-to-treat analyses were carried out for all outcomes using the Generalized Estimation Equation (GEE), to compare the outcomes between the groups (Nordic walking and free walking) and time (Pre and Post). The Bonferroni post-hoc was used to identify differences between the means for all variables. Variables with significant interactions were included in the model as covariates. We recognize that many possible factors could affect QoL, such as sex (categorical nominal variable) and cognitive function (categorical ordinal variable), and these factors were controlled as covariates in all GEE models. QoL domains were corrected by MoCA with *p* < 0.001 and by sex with *p* > 0.05. Data were analyzed using the statistical software Statistical Package for Social Science (SPSS v.20.0, Chicago, IL, USA).

## 3. Results

### 3.1. Demographic, Anthropometric, and Clinical Characteristics of the Sample

The flowchart detailing the steps of participant selection is shown in Figure 3. Baseline demographic, anthropometric, and clinical characteristics of the sample, as well as all outcome variables, are described in Table 1. Approximately 82% of the Nordic walking group participants and 75% of the free walking group participants had comparatively mild PD, modified Hoehn and Yahr stage 1–2.5, whereas 18% were in stage 3, and 25% were in stage 4, which represents moderate to severe disease. The 33 participants (13 women and 20 men) had a mean age of 72 ± 8.5 years. The Nordic walking group was taller and heavier than the free walking group. The cognitive impairment assessed by MoCA shows a worse result for Nordic walking in comparison to free walking at baseline, but these values indicate that both groups initially had mild cognitive impairment.

### 3.2. Adherence

Adherence was defined as the percentage of intervention sessions fully accomplished without protocol deviations given the total number of scheduled sessions. The program adherence was high (total completers of the Nordic walking program = 14, 87.5%, and of the free walking = 15, 88.2%). All patients who completed the intervention had 100% class attendance, demonstrating adherence to training.

### 3.3. Quality of Life

The detailed results for QoL WHOQOL OLD and WHOQOL BREF can be seen in Table 2. Regarding the domains of QoL analyzed through WHOQOL OLD, significant differences were found between the groups in the QoL general, Sensory ability, Autonomy, Past activities/present/future, Social participation, Death and dying, and Intimacy domains, with *p* < 0.001, for all. Only the autonomy, past activities/present/future and death and dying domains had statistically significant difference in the interaction between group × time (*p* = 0.05). The Bonferroni post hoc was applied, for which the means of each group were observed at each time point to identify where the difference occurred.

The Bonferroni post-hoc test for interaction time*group shows that Nordic walking was more efficient than free walking, in the field of autonomy. After the training protocol, the participants in the Nordic walking group improved and increased values of autonomy (*p* < 0.001). The same pattern occurred for the intimacy domain, adjusted by gender (*p* = 0.013). The Nordic walking group significantly increased values of intimacy after the training program (*p* = 0.001), while the free walking group reduced these values.

Regarding the Past activities/present/future domain, the Bonferroni post-hoc test for interaction time*group showed that Nordic walking and free walking groups were significantly different, with higher values for the Nordic walking group (*p* = *0*.025). Interestingly, regarding the Death and dying domain, the groups exhibited different behaviors: the free walking group showed an increase (pre-intervention = 47.34, and post-intervention = 52.00), while the Nordic walking group showed a decrease after nine weeks (pre-intervention = 65.94, and post-intervention = 52.41).

In addition, social participation revealed a statistically significant difference in relation to the group*time interaction (*p* = 0.007), and higher values were found in the Nordic walking group compared with the free walking group.

The detailed results for QoL by WHOQOL BREF can be seen in Table 2. Both groups demonstrated significant improvements in the QoL general, physical, and psychological domains of the WHOQOL-BREF after the intervention period, compared with pre-intervention (*p* < 0.05). Regarding the Environmental and Social relationship domains, higher values were found in the Nordic walking group. However, these parameters remained unchanged post-intervention in the Nordic and free walking groups, and no significant time*group interactions were found.

### 3.4. Non-Motor Symptoms

In relation to non-motor symptoms, a significant time effect was found for cognitive function (*p* = 0.046) and depressive symptoms (*p* < 0.001) post-intervention for both groups. The Nordic walking group showed higher values for cognitive function assessed by the MoCA and lower scores for depressive symptoms assessed by the GDS-15 compared with the free walking group. However, no statistically significant time*group interactions were found for these outcomes. Detailed results of cognitive assessments and depressive symptoms can be seen in Table 3.

### 3.5. Motor Symptoms

Regarding motor symptoms of PD (Table 3), there was some improvement after the exercise intervention in both groups (*p* < 0.001), although not statistically significant between the groups (*p* = 0.128). The Nordic walking group showed a decrease in motor symptoms 15.10 ± 3.2 to 11.64 ± 2.1, and the free walking group decreased from 23.19 ± 3.9 to 17.43 ± 3.8 in the UPDRS, without group*time interaction.

## 4. Discussion

The aim of this study was to examine the effects of nine-week Nordic walking and free walking training programs on QoL, non-motor symptoms (cognitive function, depressive symptoms), and motor symptoms in people with PD. The present study showed that Nordic walking seems to be more efficient than free walking in the autonomy, social participation, and intimacy domains of QoL. In addition, in the Past activities/present/future domain, the Nordic walking group had higher values compared with the free walking group post the nine-week intervention. In relation to the Death and dying domain, the free walking group showed increasing scores, while the Nordic walking scores decreased after nine weeks of training. The QoL general, physical, and psychological domains of the WHOQOL-BREF, and non-motor symptoms improved after both interventions.

Our hypothesis was that Nordic walking can be more effective than free walking in improving the QoL and non-motor symptoms in PD because of the stimulation of mechanoreceptors provoked by excitatory stimuli from using poles, leading to an increased improvement in these outcomes. Thus, the hypothesis of the study was partially accepted.

Many possible factors can affect QoL, cognitive function, and depressive symptoms in individuals with PD [4]. Significant gains in the QoL general domain are relevant to the study population because improvements in this domain may promote benefits in autonomy, independence, and social relationships in people with PD [9]. In the present study, we showed that walking with poles seems to be more effective than walking without poles for this outcome, which was shown by gains in the autonomy, social participation, and intimacy domains of QoL. Although we did not find a superiority of Nordic walking for non-motor symptoms, our periodized and individualized Nordic and free walking training programs were able to improve cognitive function and depressive and motor symptoms in this population. In general, we believe that motor-symptoms need a longer time to change. However, from a clinical point of view, our results are promising since PD is a chronic and degenerative disease.

Our findings corroborate those of previous studies showing the beneficial effects of endurance training for different purposes in people with PD [10,11,21]. QoL is an integrative evaluation of wellness and is related to general health. We observed significant effects of exercise on cognitive function, depressive symptoms, and all domains of QoL. The literature reports that endurance training is effective in improving cognitive function [22], and depressive symptoms [6]. Clinically, an improvement in cognitive function implies mitigation of harmful motor effects and non-motor symptoms, since the execution of movements to perform daily life activities requires cognition, especially executive function and attention [23,24]. Physiologically, in addition to dopaminergic alterations, changes in serotonin, cholinergic, and noradrenergic neurotransmitters are found and are causally related to cognitive and behavioral impairments [8].

The improvement in QoL (including autonomy), cognitive symptoms, and depressive symptoms in endurance/walking studies is recognized in individuals with PD, but the question in this study was whether walking with walking sticks could offer greater advantages in these outcomes compared with free walking in this population. Although the improvement in autonomy after a walking program seems obvious, the comparison of the use or not of poles during walking, until now, has not been tested in people with mild to severe Parkinson’s disease. Our results have demonstrated that Nordic walking seems to promote more significant benefits in the autonomy domain compared with free walking. This finding could be potentially explained by the adjustments necessary for the control of gait related to the central pattern generator. According to this model, other cortical adjustments are needed due to the greater complexity of the technical task related to Nordic walking, in order to sustain and support poles on the ground [13,25,26,27]. Probably the neural circuitry from other brain regions, such as the pre-motor cortex may have contributed to these findings. Consistent with our previous observation showing improvements in functional mobility [12,27] and balance [15], autonomy was enhanced using poles in comparison to free walking. These findings are valuable and collectively indicate and reinforce the benefit of Nordic walking as a particularly powerful method of physical activity and therapy.

The importance of exercise programs with progressively increasing intensity was also evidenced by the present findings, for both walking protocols. Furthermore, our findings are in line with previous results in other disorders such as fibromyalgia. Exercise at high intensity (between 60 and 80% of maximal heart rate) in people with PD seems to promote an endogenous increase of neurotrophic factors such as brain-derived neurotrophic factor (BDNF), glial derived neurotrophic factor (GDNF), insulin-like growth factor-3 (IGF3) and neurotransmitters such as dopamine and serotonin. These markers are some of the potential mechanisms for improvement of the autonomy domain in PD. To the best of our knowledge, the present study is the first to provide information related to the effects of Nordic walking in comparison to free walking on depressive symptoms and QoL in patients with PD. However, the present study has limitations. First, the sample size was relatively small, and we were unable to stratify randomization by gender, age, time, and stage of PD. Although there were no statistically significant differences between the Nordic walking and free walking groups for these variables, they may have affected the results of cognitive function and QoL as the patients had different values between the groups at baseline. These differences, however, were mitigated by adjusting the GEE models for age and sex. Second, there was no control group, so, we cannot calculate the effects on exercise associated with environmental factors related to social attention and care, in terms of social participation, e.g., Hawthorne Effect. On the other hand, using the present design, we managed to explore the differences between a widely used training method (FW) and a new method (NW). We aimed to perform a clinical trial testing the superiority of a relatively new endurance modality (NW) against a well-known endurance activity (FW) that has already shown positive results in the literature [28,29,30]. Third, the limitations include the short time of the intervention. Further research is required to determine the extent to which manipulating the training load, the training volume or the weekly frequency can reduce depressive symptoms and increase QoL, as a fundamental aspect of the treatment of individuals with PD.

The strengths of this clinical trial are the originality of the topic addressed and the promising results obtained with the two interventions, i.e., walking with and without poles, for subjects with PD with scores between 1–4 on the H&Y scale (subjects with mild to severe PD), especially regarding QoL and non-motor symptoms. Because the progression of PD symptoms leads to reduced autonomy and greater functional dependence, applying these findings in a clinical setting would be of considerable benefit to this population.

## 5. Conclusions

In conclusion, the present study reveals that nine weeks of Nordic walking training was able to improve QoL in people with PD. Our findings have demonstrated that Nordic walking promoted more significant benefits in the autonomy, intimacy, and social participation domains of QoL than free walking. Results indicated that nine-week periodized programs with moderate intensity of Nordic and free walking was sufficient to produce, similarly, improvements in non-motor symptoms in people with mild to moderate PD. In addition, Nordic walking and free walking reduced depressive symptoms in people with PD. Cognitive function was enhanced similarly in the Nordic walking and free walking groups. These results strengthen the applied and clinical potential of Nordic walking in patients with PD.

## Figures and Tables

**Figure 1 jfmk-05-00082-f001:**
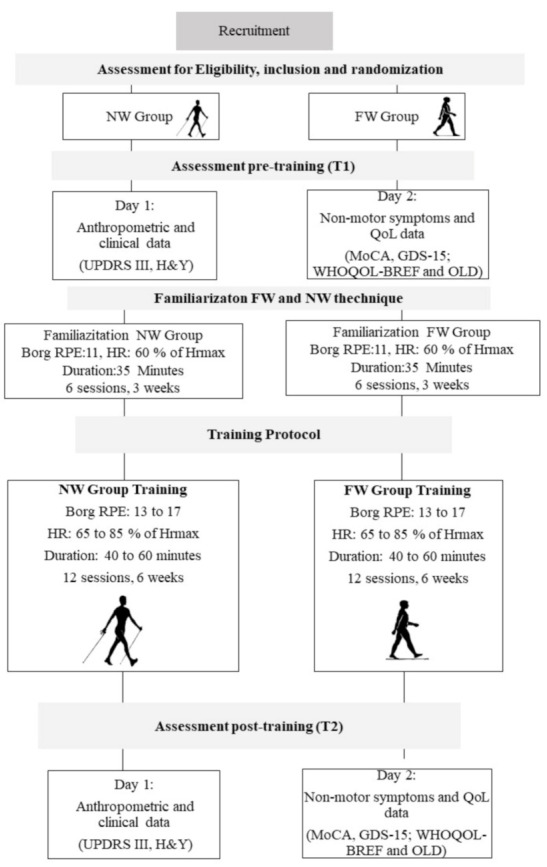
Timeline of the study. NW: Nordic walking, FW: Free walking, MoCA: Montreal cognitive assessment, UPDRS III: Unified Parkinson Disease Rating Scale—Part Motor, H&Y: Hoenh and Yahr scale, SSWS: Self-selected walking speed, RPE: Rating of perceived exertion, HR: Heart rate.

**Figure 2 jfmk-05-00082-f002:**
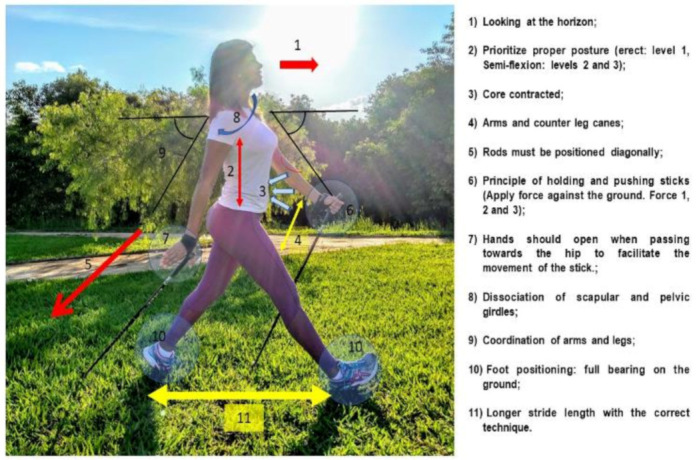
Four-support system of Nordic walking: Brazil Locomotion Method.

**Figure 3 jfmk-05-00082-f003:**
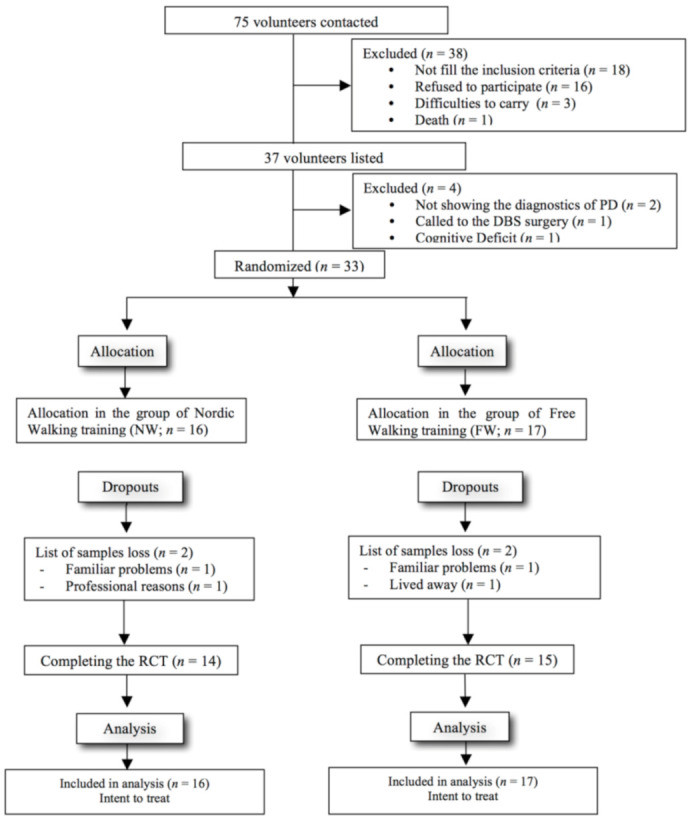
Flowchart of selection process and inclusion of volunteers. Note: Parkinson Disease (PD); Deep Brain Stimulation; Randomized clinical trial (RCT).

**Table 1 jfmk-05-00082-t001:** Variables of sample characterization for the nordic and free walking groups in baseline.

Variable	NW Group (*n* = 16)	FW Group (*n* = 17)	*p* Value
Age (years)	64.9 ± 10.2	70.5 ± 5.8	0.062
Body mass (kg)	79.0 ± 15.1	68.9 ± 11.9	0.041 *
Women, *n* (%)	3 (18)	10 (58)	0.757
Stature (m)	1.67 ± 0.08	1.59 ± 0.14	0.049 *
Body Mass Index (kg/m^2^)	28.5 ± 4.2	27.4 ± 5.8	0.556
Fat percentage (%)	21.42 ± 5.7	24.13 ± 9.3	0.367
Length of lower limb (m)	0.88 ± 0.02	0.85 ± 0.03	0.087
Time of clinical diagnostics of PD (years)	5.5 ± 3.3	5.1 ± 4.1	0.757
UPDRS III	15.00 ± 3.2	23.19 ± 3.9	0.128
Hoehn & Yahr (scale of 1 a 4)	1.5 ± 0.5	2.0 ± 1.0	0.123
MoCA (score de 0 a 30)	16.67 ± 1.4	21.96 ± 1.1	0.004 *
Berg Balance Scale	51.19 ± 1.2	47.44 ± 2.5	0.988
Clinical Symptoms	R = 7; L = 9	R = 9; L = 7	NA
Affected leg	9	6	NA
Instability (change of balance)	7	7	NA
Tremor	6	8	NA
Postural changes	14	6	NA
Rigidity	4	11	NA
Bradykinesia	4	0	NA
Dyskinesia	3	2	NA
*Freezing*	5	5	NA
Historic of falls			
Drugs (Dosages)	250 ± 0.0	181.89 ± 46.81	NA
Levodopa + Carbidopa	161 ± 53.21	250 ± 108.40	NA
Prolopa	150 ± 93.24	200 ± 0.0	NA
Sifrol	-	400 ± 0.0	NA
Biperideno	-	175 ± 75.00	NA
Benserazida	-	150. ± 00	NA
Selegina			

Note: The data are in the means and standard deviations and some variables in absolute numbers. * Statistical significance at *p* < 0.05; NW: Nordic Walking; PD: Parkinson’s disease; UPDRS III: Unified Parkinson’s Disease Rating Scale; MoCA: Montreal Cognitive Assessment; R (right); L (Left); NA (Not Apply).

**Table 2 jfmk-05-00082-t002:** Means and 95% confidence interval to WHOQOL OLD and WHOQOL BREF (dependent variables) for Nordic and Free Walking (groups and 2 moments—pre and post—of data collection; independent variables).

		PRE	POST	*p* value		
Domains of QoL	Intervention	Mean (CI 95%)	Mean (CI 95%)	Group	Time	Time × Group
***WHOQOL OLD***						
QoL general	FW	53.97	55.76	<0.001 *	0.233	0.989
	NW	(41.88; 57.89) 67.89	(47.04; 60.56) 69.71			
Sensory ability	FW	(59.34; 73.11) 56.36 (48.39; 64.32)	(63.34; 78.84) 59.16 (52.68; 65.63)	0.001 *	0.982	0.536
	NW	69.25 (62.26; 76.23)	66.64 (58.97; 74.32)			
Autonomy	FW	51.96 (47.33; 56.58)	50.17 (40.86; 59.41)	<0.001 *	0.131	0.033 *
	NW	64.38 (58.32; 70.44)	75.32 (68.76; 81.88)			
Past activities/present/future	FW	69.46 (46.37; 74.32)	63.41 (49.69; 75.87)	0.025 *	0.091	0.045 *
	NW	71.18 (68.39; 73.79)	72.74 (71.48; 82.50)			
Social participation	FW	50.53 (44.26; 56.80)	57.38 (53.06; 61.70)	<0.001 *	0.007 *	0.934
	NW	61.63 (56.35; 66.92)	68.92 (63.52; 74.33)			
Death and dying	FW	47.34 (40.48; 54.21)	52.00 (43.78; 60.22)	0.039 *	0.325	0.047 *
	NW	65.94 (57.15; 74.55)	52.41 (41.02; 63.08)			
Intimacy	FW	48.57 (42.71; 54.43)	52.61 (45.65; 59.58)	<0.001 *	0.033 *	0.544
	NW	76.20 (69.50; 82.90)	83.50 (75.81; 91.18)			
***WHOQOL BREF***						
QoL general	FW	50.06 (43.16; 56.97)	63.70 (55.73; 71.67)	0.002 *	<0.001 *	0.629
	NW	66.30 (59.81; 72.79)	76.88 (71.25; 82.50)			
Physical	FW	51.63 (46.69; 56.57)	54.80 (49.04; 60.56)	0.003 *	0.037 *	0.453
	NW	60.36 (54.43; 66.28)	67.23 (61.34; 73.11)			
Psychological	FW	57.53 (51.88; 63.19)	58.65 (54.22; 63.09)	0.006 *	0.019 *	0.093
	NW	67.65 (61.16; 74;14)	75.02 (66.58; 83.47)			
Environmental	FW	60.26 (54.71; 65.80)	61.81 (57.74; 65.89)	0.025 *	0.091	0.313
	NW	65.71 (60.32; 69.96)	71.39 (65.76; 77.03)			
Social relationship	FW	58.28 (52.75; 63.82)	59.72 (55.44; 63.99)	<0.001 *	0.216	0.433
	NW	65.63 (61.11; 70.16)	72.05 (66.90; 77.19)			

Note: * Statistical significance at *p* < 0.05; CI: confidence interval; WHOQOL (World Health Organization Instrument for Quality of Life Assessment); QoL (Quality of Life); PRE (initial evaluation before training); POST (evaluation after training); FW (Free Walking); NW (Nordic Walking). The domains of social participation and intimacy were corrected by sex (*p* = 0.05 and *p* = 0.013, respectively); the other domains of QoL were corrected by MoCA with *p* < 0.001 and by sex with *p* < 0.05.

**Table 3 jfmk-05-00082-t003:** Means and standard error to cognitive function, depressive symptoms, motor symptoms of disease for nordic and free walking groups in two different times.

		PRE	POST			
Variable	Intervention	Means ± SE	Means ± SE	Group	Time	Group × Time
Cognitive function (MoCA)	FW	16.06 ± 1.1	17.29 ± 1.7	0.004 *	0.046 *	0.784
	NW	21.50 ± 1.1	22.43 ± 1.2			
Depressive symptoms (GDS—15)	FW	4.6 ± 0.4	3.7 ± 0.4	0.014 *	<0.001 *	0.583
	NW	2.8 ± 0.3	1.2 ± 0.3			
Motor Symptoms (UPDRS III)	FW	23.19 ± 3.9	17.43 ± 3.8	0.128	<0.001 *	0.352
	NW	15.10± 3.2	11.64 ± 2.1			

* Significant at *p* < 0.05. Abbreviations: PRE, initial evaluation before training + before familiarization; POST, evaluation after training; SE: standard error; The Cognitive function was evaluated by MoCA: Montreal Cognitive Assessment; The depressive symptoms was assessed by GDS-15: Geriatric Depression Scale—15 items; The motor symptoms was assessed by UPDRS III: Unified Parkinson’s Disease Rating Scale; FW, free walking; NW, nordic walking. Note: the depressive symptoms were corrected by MoCA (*p* < 0.001) and by sex (*p* < 0.05) such as covariates.

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
