# Peer review of "Nordic Walking and Free Walking Improve the Quality of Life, Cognitive Function, and Depressive Symptoms in Individuals with Parkinson’s Disease: A Randomized Clinical Trial"

_jfmk, 2020, doi:10.3390/jfmk5040082_

Round 1
Reviewer 1 Report
- Could you explain the independent and dependent variables you chose in the GEE model in Table 2? Since many possible factors could affect quality of life, these factors should be controlled in the GEE models to avoid the biased estimation. However, I could not find any clear description about the construction of the GEE model. In addition, you mentioned in the note of the Table 2 that “The domains of social participation and intimacy were corrected by sex (P=0.05 and P=0.013, respectively); the other domains of QoL were corrected by MoCA with P<0.001 and by sex with P<0.05.” Does it indicate that you put different independent variables in the GEE models when assessing different domain of QoL?
- In the discussion, you mentioned that both Nordic walking and free walking could improve QoL, cognition and depressive symptoms of patient with PD. I would favor that your study results were not sufficient to support this conclusion. This is because the primary outcome of your study was aimed to compared QoL and non-motor symptoms between different exercise training program. Your team did not included patient without receiving exercise as the control group. You could not exclude the possibility that the improvement in these outcome variables might be attributable to other environmental factors after joining the RCT program, such as the close contact with medical affairs. Therefore, the evidence from your study results to support your conclusion is also not solid.
Author Response
Dear Editor-in-chief Ph.D. Ms. Molly Lu,
Dear esteemed reviewers,
Manuscript Title: “Nordic walking and free walking improve the quality of life, cognitive function, and depressive symptoms in individuals with Parkinson's disease: a randomized clinical trial”.
Thank you for the prompt review of our manuscript. We are pleased that all reviewers noted the importance of our work and offered additional, valuable suggestions to further improve the quality of our manuscript.
In our resubmitted version, we have tried to accommodate the concerns as much as possible. All changes in the revised manuscript are typed in red and "Track Changes" function in Microsoft Word. The manuscript listed below was edited for proper English language, grammar, punctuation, spelling, and overall style by an expert English editor at Workout Translate.
We have addressed these comments in full and answered every suggestion and comment in red. We look forward to hearing from you soon regarding the outcome of our manuscript.
Yours sincerely,
Passos-Monteiro, and co-authors.
Response to Reviewer 1 Comments
- Could you explain the independent and dependent variables you chose in the GEE model in Table 2? Since many possible factors could affect the quality of life, these factors should be controlled in the GEE models to avoid biased estimation. However, I could not find any clear description of the construction of the GEE model. In addition, you mentioned in the note of Table 2 that “The domains of social participation and intimacy were corrected by sex (P=0.05 and P=0.013, respectively); the other domains of QoL were corrected by MoCA with P<0.001 and by sex with P<0.05.” Does it indicate that you put different independent variables in the GEE models when assessing different domains of QoL?
RESPONSE: Thank you for this question. Quality of life outcomes and non-motors parameters (cognition and depressive symptoms) were compared between the Nordic Walking and Free Walking, time (pre- and post-intervention), and group*time interaction (We added this information in lines 158-160).
In relation to GEE, considering the literature, we believe that the GEE method is the most appropriate method to analyze the data of the present study which characterizes a longitudinal design with paired samples (Liang KY, Zeger SL. Longitudinal data analysis using generalized linear models. Biometrika, 1986;73:13-22).
We chose the Generalized Estimating Equations (GEE) method, to better applied the correct principles of analysis of longitudinal data in clinical trials. Considering that this method was designed to analyze paired and longitudinal data (Liang & Zeger, 1986), and that we had two independents factors (2 groups – Nordic Walking and Free walking, and 2 moments - pre and post - of data collection). In relation to dependent variables, we choose all domains of WHOQOL OLD and WHOQOL BREF. We have clarified this information on the manuscript, including this information in table 2: “Table 2. Means and 95% confidence interval to WHOQOL OLD and WHOQOL BREF (dependent variables) for Nordic and Free Walking (groups and 2 moments - pre and post - of data collection: independent variables). We chose not to apply other Statistic Analysis, like as the Analysis of Variance (ANOVA), for the following reasons:
1) Before the data collection, we could not know if we would have any variable with other than “normal” distribution. The GEE can be applied regardless of the data distribution.
2) The assumption of sphericity between all the different moments required by the ANOVA test is exceedingly difficult to achieve in studies with health outcomes in humans, especially in clinical populations, like our study. And the GEE method does not have this requirement.
Yes, we recognize that many possible factors could affect the quality of life like as sex (categorical nominal variable), cognition (categorical ordinal variable) and these factors were controlled such as co-variates in all GEE models (we added this information in the lines 221-223). In addition, these variables were included in the model as covariates because they have significant interactions.
- In the discussion, you mentioned that both Nordic walking and free walking could improve QoL, cognition, and depressive symptoms of the patient with PD. I would favor that your study results were not sufficient to support this conclusion. This is because the primary outcome of your study was aimed to compared QoL and non-motor symptoms between different exercise training programs. Your team did not include patient without receiving exercise as the control group. You could not exclude the possibility that the improvement in these outcome variables might be attributable to other environmental factors after joining the RCT program, such as the close contact with medical affairs. Therefore, the evidence from your study results to support your conclusion is also not solid.
Thank you for this comment. Although we could not rule the non-specific (environmental effects) of both NW or FW in our study on cognitive and depressive symptoms, these environmental factors are also present in any supervised exercise interventions and, yet they could not be attributable to the specific biological effects of exercise (such as physiological factors), they can be attributable for the intervention as a whole. We have now added to the limitations of the following statement. We recognize this limitation add this information in lines 385-396:
Second, there was no control group, so, we cannot calculate the amount of the effects of exercise-associated to environmental factors, related to social attention, care, in terms of social participation, e.g., Hawthorne Effect. On the other hand, using the present design, we managed to explore the differences between a widely used training method (FW) and a new method (NW). Also, we aimed to perform a clinical trial testing the superiority of a relatively new endurance modality (Nordic walking) against a well-known endurance activity (free walking) which the literature has already been demonstrating positive results. Third, the limitations include the short time of the intervention.
Thank you for your point of view in relation to our conclusion, we have clarified the information (439-447):
In conclusion, the present study reveals that nine-week of Nordic walking training was able to improve the QoL on PD. Our findings have demonstrated that Nordic walking promoted more significant benefits on autonomy, intimacy, and social participation domains of QoL than the free walking. The results indicate that a nine-week periodized program with a moderate intensity of Nordic and free walking was sufficient to produce, similarly, improvements in non-motor symptoms in people with mild-to-moderate PD. In addition, Nordic and free walking reduce depressive symptoms in people with PD. The cognitive function is enhanced similarly in Nordic and free walking. These results strengthen the applied and clinical potential of Nordic walking in patients with PD.
In relation to the control group, we chose to compare the superiority effects of two interventions, Nordic walking, and free walking, for ethical reasons. We add this information in lines: 107-112:
“We aimed to perform a clinical trial testing the superiority of a relatively new endurance modality (Nordic walking) against a well-known endurance activity (free walking) which the literature has already been demonstrating positive results. We chose to compare the superiority effects of two interventions, Nordic walking, and free walking, for ethical reasons. Physical exercise is extremely important in maintaining and improving symptoms in people with Parkinson's disease.”
We are very grateful for your suggestions and comments, which have contributed to the scientific quality of the study. We have carefully reviewed each point and hope that we have managed to meet your expectations in this review.

Reviewer 2 Report
The study and the results are respectively well designed and presented. Therefore, according to me, it is necessary to proceed just with minor revisions:
- please carefully check the manuscript for typos (e.g. line 84)
- please improve non-motor and motor symptoms results. The explanations are not clear
- please carefully check the tables for proper presentation of results (e.g. when you write that data are presented as MEAN±DP (what about DP?), are the data presented as MEAN±DP? For example, check table 3. In addition, why in table 2, in autonomy line, do you write data as MEAN±DP? What about NA in table 1? Please insert in the legend and explain its significance)
- eligibility criteria: what about being sedentary? How did you state this condition? What about the cutoff to consider a person sedentary or not?
- exercise training. Lines 165-169. What about the content of the point 2 of the workout session? What about Locomotion Method Brazil? Please clarify and carefully describe. What about the part 2 of the free Walking group? Please carefully describe.
- You stated that Nordic Walking improved autonomy better than free walking. Could you better explain? Don't you consider this result obvious when you train for a task (with specific exercise) and not?
- What about the fact that while autonomy showed a groupxtime effect non-motor and motor symptoms not? could you insert your opinion about that?
- according to me, appendices and videos are not necessary. I suggest to eliminate them. On the contrary, did you respect all the procedures concerning the authorizations to publish photos/videos with recognizable faces?
Author Response
Dear Editor-in-chief Ph.D. Ms. Molly Lu,
Dear esteemed reviewers,
Manuscript Title: “Nordic walking and free walking improve the quality of life, cognitive function, and depressive symptoms in individuals with Parkinson's disease: a randomized clinical trial”.
Thank you for the prompt review of our manuscript. We are pleased that all reviewers noted the importance of our work and offered additional, valuable suggestions to further improve the quality of our manuscript.
In our resubmitted version, we have tried to accommodate the concerns as much as possible. All changes in the revised manuscript are typed in red and "Track Changes" function in Microsoft Word. The manuscript listed below was edited for proper English language, grammar, punctuation, spelling, and overall style by an expert English editor at Workout Translate.
We have addressed these comments in full and answered every suggestion and comment in red. We look forward to hearing from you soon regarding the outcome of our manuscript.
Yours sincerely,
Passos-Monteiro, and co-authors.
Response to Reviewer 2 Comments
The study and the results are respectively well designed and presented. Therefore, according to me, it is necessary to proceed just with minor revisions:
RESPONSE: Thank you for the prompt review of our manuscript. We are pleased that you noted the importance of our work and offered additional, valuable suggestions to further improve the quality of our manuscript.
- please carefully check the manuscript for typos (e.g. line 84)
RESPONSE: Thank you for your observation. We carefully this sentence in lines 83-84:
“no data have been reported about the effects of the Nordic walking training program in comparison to a free walking training program on QoL and non-motor symptoms of the peoples with PD”.
- please improve non-motor and motor symptoms results. The explanations are not clear
Lines 151 -169:
Quality of life outcomes and non-motors parameters (cognitive function and depressive symptoms) were compared between the Nordic Walking and Free Walking, time (pre- and post-intervention), and group*time interaction. The QoL was assessed with the following instruments: the questionnaire World Health Organization Instrument for Quality of Life Assessment (WHOQOL-BREF domains: physical, psychological, social relationships, environment, and general quality of life) and (WHOQOL-OLD domains: sensory abilities, autonomy, ‘Past, Present and Future Activities’, social participation, death and dying, intimacy, and general quality of life).[29].
The non-motor symptoms (cognitive function, depressive symptoms) were assessed. The Montreal Cognitive Assessment (MoCA), is a brief screening tool for mild cognitive impairment. This evaluation accesses different cognitive domains and investigates the individual's abilities in the following areas: attention and concentration, executive functions, memory, language, visuoconstructive skills, conceptualization, calculation, and orientation. The total score of the MoCA is 30 points, with a score of 26, or more, considered normal and less than 26 is considered a cognitive impairment. The depressive symptoms were measured by Geriatric Depression Scale - 15 items (GDS-15) The scale consists of 15 dichotomous questions in which participants are asked to answer yes or no about how they felt over the past week (for instance, “Do the patient feel that their life is empty?,” Do the patient feel that their situation is hopeless?). Scores range from 0 to 15 with higher scores indicating more depressive symptoms.
- please carefully check the tables for proper presentation of results (e.g. when you write that data are presented as MEAN±DP (what about DP?), are the data presented as MEAN±DP? For example, check table 3. In addition, why in table 2, in the autonomy line, do you write data as MEAN±DP? What about NA in table 1? Please insert in the legend and explain its significance)
RESPONSE: Thank you for your comments, you are right. We apologize for the mistake. We have added legend and corrected all this information in tables 1,2 and 3. In Table 1 the data are presented in mean and standard deviation (ANOVA), in table 2 we choose presented the data in mean and confidence Interval, and in table 3 in the mean and standard error of the mean (GEE).
|
Table 1. Variables of sample characterization for the nordic and free walking groups in baseline |
|||
|
Variable |
NW group (n =16) |
FW group (n =17) |
P value |
|
Age (years) |
64.9 ± 10.2 |
70.5 ± 5.8 |
0.062 |
|
Body mass (kg) |
79.0 ± 15.1 |
68.9 ± 11.9 |
0.041* |
|
Women, n (%) |
3 (18) |
10 (58) |
0.757 |
|
Stature (m) |
1.6 ± 0.0 |
1.5 ± 0.1 |
0.049* |
|
Body Mass Index (kg/m²) |
28.5 ± 4.2 |
27.4 ± 5.8 |
0.556 |
|
Fat percentage (%) Length of lower limb (m) |
21.42 ± 5.7 0.88 ± 0.0 |
24.13 ± 9.3 0.85 ± 0.0 |
0.367 0.087 |
|
Time of clinical diagnostics of PD (years) |
5.5 ± 3.3 |
5.09 ± 4.1 |
0.757 |
|
UPDRS III |
15.00 ± 3.2 |
23.19 ± 3.9 |
0.128 |
|
Hoehn & Yahr (scale of 1 a 4) |
1.5 ± 0 .5 |
2.0 ± 1.0 |
0.123 |
|
MoCA (score de 0 a 30) |
16.67 ± 1.4 |
21.96 ± 1.1 |
0.004* |
|
Berg Balance Scale Clinical Symptoms Affected leg Instability (change of balance) Tremor Postural changes Rigidity Bradykinesia Dyskinesia Freezing Historic of falls Drugs (Dosages) Levodopa + Carbidopa Prolopa Sifrol Biperideno Benserazida Selegina |
51.19 ± 1.2 R =7; L = 9 9 7 6 14 4 4 3 5
250 ± 0.0 161 ± 53.21 150 ± 93.24 - - - |
47.44 ± 2.5 R =9; L = 7 6 7 8 6 11 0 2 5
181.89 ± 46.81 250 ± 108.40 200 ± 0.0 400 ± 0.0 175 ± 75.00 150. ± 00 |
0.988 NA NA NA NA NA NA NA NA NA
NA NA NA NA NA NA |
Note: The data are in the means and standard deviations and some variables in absolute numbers. * Statistical significance at P<0.05; NW: Nordic Walking; PD: Parkinson’s disease; UPDRS III: Unified Parkinson’s Disease Rating Scale; MoCA: Montreal Cognitive Assessment; R (right); L (Left); NA (Not Apply).
|
Table 2. Means and 95% confidence interval to WHOQOL OLD and WHOQOL BREF (dependent variables) for Nordic and Free Walking (groups and 2 moments - pre and post - of data collection; independent variables). |
|||||||
|
|
PRE |
POST |
P value |
||||
|
Domains of QoL (WHOQOL OLD) |
Intervention |
Mean (CI 95%) |
Mean (CI 95%) |
Group |
Time |
Time*Group |
|
|
QoL general |
FW |
53.97 |
55.76 |
<0.001* |
0.233 |
0.989 |
|
|
NW |
(41.88; 57.89) 67.89 |
(47.04; 60.56) 69.71 |
|||||
|
Sensory ability |
FW |
(59.34; 73.11) 56.36 (48.39; 64.32) |
(63.34; 78.84) 59.16 (52.68; 65.63) |
0.001* |
0.982 |
0.536 |
|
|
NW |
69.25 (62.26; 76.23) |
66.64 (58.97; 74.32) |
|||||
|
Autonomy |
FW |
51.96 (47.33; 56.58) |
50.17 (40.86; 59.41) |
<0.001* |
0.131 |
0.033* |
|
|
NW |
64.38 (58.32; 70.44) |
75.32 (68.76; 81.88) |
|||||
|
Past activities/present/future |
FW |
69.46 (46.37; 74.32) |
63.41 (49.69; 75.87) |
0.025* |
0.091 |
0.045* |
|
|
NW |
71.18 (68.39; 73.79) |
72.74 (71.48; 82.50) |
|||||
|
Social participation |
FW |
50.53 (44.26; 56.80) |
57.38 (53.06; 61.70) |
<0.001* |
0.007* |
0.934 |
|
|
NW |
61.63 (56.35; 66.92) |
68.92 (63.52; 74.33) |
|||||
|
Death and dying |
FW |
47.34 (40.48; 54.21) |
52.00 (43.78; 60.22) |
0.039* |
0.325 |
0.047* |
|
|
NW |
65.94 (57.15; 74.55) |
52.41 (41.02; 63.08) |
|||||
|
Intimacy |
FW |
48.57 (42.71; 54.43) |
52.61 (45.65; 59.58) |
<0.001* |
0.033* |
0.544 |
|
|
NW |
76.20 (69.50; 82.90) |
83.50 (75.81; 91.18) |
|||||
Domains of QoL WHOQOL BREF
|
QoL general |
FW |
50.06 (43.16; 56.97) |
63.70 (55.73; 71.67) |
0.002* |
<0.001* |
0.629 |
|
NW |
66.30 (59.81; 72.79) |
76.88 (71.25; 82.50) |
||||
|
Physical |
FW |
51.63 (46.69; 56.57) |
54.80 (49.04; 60.56) |
0.003* |
0.037* |
0.453 |
|
NW |
60.36 (54.43; 66.28) |
67.23 (61.34; 73.11) |
||||
|
Psychological |
FW |
57.53 (51.88; 63.19) |
58.65 (54.22; 63.09) |
0.006* |
0.019* |
0.093 |
|
NW |
67.65 (61.16; 74;14) |
75.02 (66.58; 83.47) |
||||
|
Environmental |
FW |
60.26 (54.71; 65.80) |
61.81 (57.74; 65.89) |
0.025* |
0.091 |
0.313 |
|
NW |
65.71 (60.32; 69.96) |
71.39 (65.76; 77.03) |
||||
|
Social relationship |
FW |
58.28 (52.75; 63.82) |
59.72 (55.44; 63.99) |
<0.001* |
0.216 |
0.433 |
|
NW |
65.63 (61.11; 70.16) |
72.05 (66.90; 77.19) |
||||
|
Note: * Statistical significance at P<0.05; CI: confidence interval; WHOQOL (World Health Organization Instrument for Quality of Life Assessment); QoL (Quality of Life); PRE (initial evaluation before training); POST (evaluation after training); FW (Free Walking); NW (Nordic Walking). The domains of social participation and intimacy were corrected by sex (P=0.05 and P=0.013, respectively); the other domains of QoL were corrected by MoCA with P<0.001 and by sex with P<0.05. |
||||||
Table 3. Means and standard error of mean to cognitive function, depressive symptoms, motor symptoms of the disease for nordic, and free walking groups in two different times.
|
|
|
PRE |
POST |
||||||
|
Variable |
Intervention |
Mean ± SE
|
Mean ± SE Group Time Group*Time |
||||||
|
Cognitive function (MoCA) |
FW |
16.06 ± 1.1 |
17.29 ± 1.7 |
0.004* |
0.046* |
0.784 |
|
||
|
NW |
21.50 ± 1.1 |
22.43 ± 1.2 |
|
||||||
|
Depressive symptoms (GDS – 15) |
FW |
4.6 ± 0.4 |
3.7 ± 0.4 |
0.014* |
< 0.001* |
0.583 |
|
||
|
NW |
2.8 ± 0.3 |
1.2 ± 0.3 |
|
||||||
|
Motor Symptoms (UPDRS III) |
FW |
23.19 ± 3.9 |
17.43 ± 3.8 |
0.128 |
<0.001* |
0.352 |
|
||
|
NW |
15.10± 3.2 |
11.64 ± 2.1 |
|
||||||
|
|
|
|
|
||||||
*Significant at P<0.05. Abbreviations: PRE, initial evaluation before training + before familiarization; POST, evaluation after training; SE: standard error; The Cognitive function was evaluated by MoCA: Montreal Cognitive Assessment; The depressive symptoms was assessed by GDS-15: Geriatric Depression Scale – 15 items; The motor symptoms was assessed by UPDRS III: Unified Parkinson’s Disease Rating Scale; FW, free walking; NW, nordic walking. Note: the depressive symptoms were corrected by MoCA (P< 0,001) and by sex (P< 0,05) such as covariates.
- eligibility criteria: what about being sedentary? How did you state this condition? What about the cutoff to consider a person sedentary or not?
RESPONSE: Thank you for these questions. We choose sedentary people with PD as eligibility criteria, for two reasons: 1) Some studies (ELLIGSON et al., 2019) shows the influence of sedentary time on factors associated with quality of life in people with PD; 2) To control and to know if the improvements in the outcomes in the present study will be from the effects of Nordic or free Walking program and not from other physical activities.
We considered sedentary, people without physical activity for at least six months, as usual in exercise studies. Thus, we have added this sentence in lines 125-126: “being sedentary or do not practice any type of physical activity for at least six months”
Ellingson LD, Zaman A, Stegemöller EL. Sedentary Behavior and Quality of Life in Individuals With Parkinson's Disease. Neurorehabil Neural Repair. 2019;33(8):595-601. doi:10.1177/1545968319856893
- exercise training. Lines 165-169. What about the content of the point 2 of the workout session? What about Locomotion Method Brazil? Please clarify and carefully describe. What about the part 2 of the free Walking group? Please carefully describe.
RESPONSE: Thank you for your comments! To clarify we added and carefully described about the workout session. We have added a paragraph about it (lines 187-195) and Nordic Walking Locomotion Brazil.
The NW and FW training protocol progressed over time until reaching 60 minutes in the last training cycle. Every three sessions had a regenerative session with duration of the 40 minutes at SSWS, stretching with poles (for NW group) and body relaxation techniques. The training protocol consisted of three parts: (1) joint mobilization and warm-up with free three-minutes’ walk at SSWS (eight min); (2) main part - specific training (35 to 50 min): continuous or interval walking on different terrains (i.e. track, grass and sand), with two-to-five minutes of rest in predetermined times, and standardized among all subjects. The difference between the workout sessions in this point, was just the use or not the poles during the walking (3) cool-down with stretching, core, and respiration exercises (five minutes). The periodization method is based in Monteiro et al. [26].
- You stated that Nordic Walking improved autonomy better than free walking. Could you better explain? Don't you consider this result obvious when you train for a task (with specific exercise) and not?
RESPONSE: Thank you for these questions. The authors understanding your point of view. However, we kindly invite you to understand our research question: The improvement in the quality of life (inclusive autonomy), cognitive symptoms, and depressive symptoms in endurance / walking studies are recognized in the peoples with PD, but the question of the study was whether walking with sticks would offer greater advantages in these outcomes in relation to the free walking group for this population. And although it seems obvious the improvement in autonomy after walking program, the comparison of the use of poles or not during the walk, until the present study had not been tested in peoples with mild-to-severe Parkinson's disease.
- What about the fact that while autonomy showed a groupxtime effect non-motor and motor symptoms not? could you insert your opinion about that?
RESPONSE: Thank you for your question. We added our view about this in lines 327-339:
Many possible factors could affect the quality of life, like as cognitive function and depressive symptoms, especially in individuals with PD [4]. Significant gains in QoL general are relevant to the study population because improvements in this regard may promote benefits in the autonomy, independence, and social relationship of peoples with PD [30]. In the present study, we showed that walking with poles can be more effective than walking without poles to this outcome, reveled for gains in the field of autonomy, social participation, and intimacy domain of QoL. Although we do not find the superiority of Nordic Walking to nom-motor symptoms, our periodized and individual of the Nordic and free walking training program, with moderate intensity during nine-week improvements considerable benefit to cognitive function, depressive and motor symptoms of this population. In general, we believed that the motor-symptoms need of a long time to higher changes and to the possible differences between nordic walking and free walking. Despite, the point of the view clinicians, our results are promising since Parkinson's disease is chronic and degenerative.
- according to me, appendices and videos are not necessary. I suggest to eliminate them. On the contrary, did you respect all the procedures concerning the authorizations to publish photos/videos with recognizable faces?
Thank you for your comments. We understand your point. However, we have decided to keep the appendices and videos because we followed all procedures concerning the ethical and authorizations to publish photos/videos with recognizable faces.
We are incredibly grateful for your suggestions and comments, which have contributed to the scientific quality of the study. We have carefully reviewed each point and hope that we have managed to meet your expectations in this review.
For this round, the reviewers and editor did not request any other points to be reviewed.
Therefore, we are incredibly grateful for all recommendations that have improved our manuscript for this moment. We look forward to hearing from you soon regarding the outcome of our manuscript.
Yours sincerely,
Passos-Monteiro and co-authors.

Round 2
Reviewer 1 Report
The manuscript could be accepted in this version.